# Comparative Study of CoFe_2_O_4_ Nanoparticles and CoFe_2_O_4_-Chitosan Composite for Congo Red and Methyl Orange Removal by Adsorption

**DOI:** 10.3390/nano11030711

**Published:** 2021-03-12

**Authors:** Claudia Maria Simonescu, Alina Tătăruş, Daniela Cristina Culiţă, Nicolae Stănică, Ioana Alexandra Ionescu, Bogdan Butoi, Ana-Maria Banici

**Affiliations:** 1Department of Analytical Chemistry and Environmental Engineering, Faculty of Applied Chemistry and Materials Science, Politehnica University of Bucharest, Polizu Street, No. 1-7, District 1, 011061 Bucharest, Romania; alina.tatarus@yahoo.com; 2National Research and Development Institute for Industrial Ecology, INCD ECOIND Bucuresti, 71-73 Drumul Podul Dambovitei Str., 060652 Bucharest, Romania; ioana.ionescu@incdecoind.ro; 3Ilie Murgulescu Institute of Physical Chemistry, 202 Splaiul Independentei, 060021 Bucharest, Romania; nstanica@icf.ro; 4National Institute for Laser, Plasma and Radiation Physics, 077125 Măgurele, Romania; bogdan.butoi@inflpr.ro (B.B.); niculescu.anam@gmail.com (A.-M.B.)

**Keywords:** dye adsorption, Congo Red, Methyl Orange, magnetic adsorbents, isothermal study, kinetic study

## Abstract

(1) Background: A comparative research study to remove Congo Red (CR) and Methyl Orange (MO) from single and binary solutions by adsorption onto cobalt ferrite (CoFe_2_O_4_) and cobalt ferrite–chitosan composite (CoFe_2_O_4_-Chit) prepared by a simple coprecipitation method has been performed. (2) Methods: Structural, textural, morphology, and magnetic properties of the obtained magnetic materials were examined by X-ray diffraction (XRD), Fourier-transform infrared (FTIR) spectroscopy, N_2_ adsorption–desorption analysis, Scanning Electron Microscopy (SEM), Transmission Electron Microscopy (TEM), and magnetic measurements. The optimal operating conditions of the CR and MO removal processes were established in batch experiments. The mathematical models used to describe the processes at equilibrium were Freundlich and Langmuir adsorption isotherms. (3) Results: Cobalt ferrite–chitosan composite has a lower specific surface area (S_BET_) and consequently a lower adsorption capacity than cobalt ferrite. CoFe_2_O_4_ and CoFe_2_O_4_–Chit particles exhibited a superparamagnetic behavior which enabled their efficient magnetic separation after the adsorption process. The research indicates that CR and MO adsorption onto prepared magnetic materials takes place as monolayer onto a homogeneous surface. According to Langmuir isotherm model that best fits the experimental data, the maximum CR/MO adsorption capacity is 162.68/94.46 mg/g for CoFe_2_O_4_ and 15.60/66.18 mg/g for CoFe_2_O_4_–Chit in single solutions. The results of the kinetics study revealed that in single-component solutions, both pseudo-first-order and pseudo-second-order kinetics models represent well the adsorption process of CR/MO on both magnetic adsorbents. In binary solutions, adsorption of CR/MO on CoFe_2_O_4_ better follows the pseudo-second-order kinetics model, while the kinetic of CR/MO adsorption on CoFe_2_O_4_–Chit is similar to that of the dyes in single-component solutions. Acetone and ethanol were successfully used as desorbing agents. (4) Conclusions: Our study revealed that CoFe_2_O_4_ and CoFe_2_O_4_–Chit particles are good candidates for dye-contaminated wastewater remediation.

## 1. Introduction

Improper disposal of pollutants such as heavy metal ions, dyes, pharmaceutical wastes, pesticides, organic compounds into aquatic environments constitutes one of the significant environmental issues that is facing the entire world [1]. Dyes are basic chemical compounds that are utilized in many industrial practices such as textile, paper, leather, plastics and rubber, food, pharmaceutical, and cosmetics industrial practices. At the industrial level, more than 10,000 dyes and pigments are used in various fields [2]. Dyes are intensely colored substances that present special problems in wastewater, even in very small quantities. However, the effect is more unpleasant from an aesthetic point of view, rather than dangerous, for example, red dyes spilled into rivers and oceans.

Dye molecules are mutagenic, carcinogenic, and cause dysfunction in the following organs: the kidneys, liver, brain, and reproductive and central nervous system [2]. When effluents with dyes are discharged into various water sources, the turbidity of the water increases as the effluents will produce a visible layer on the surface of the water sources due to the low density (0.8 kg/m^3^) compared to the density of water (1 kg/m^3^) [3]. The sunlight necessary for the processes of photosynthesis and respiration of aquatic organisms will not penetrate through the dye layer determining the death or aquatic organisms. The decrease in soil productivity can be registered in the situation when effluents containing dyes will be discharged in soils by blocking their pores [3]. 

Dyes used in the textile industry have negative effects on the DNA in cells, raising the risk of cancer. Those used in the food industry are associated with the appearance of hives, allergies, asthma, hyperactivity, irritability, aggression, dermatitis, and angioedema [4], and those used in the pharmaceutical industry are known to cause skin irritations. Bladder, blood and bone marrow cancer, fertility problems, and allergies can be caused by dyes in the cosmetics industry [4]. Cationic and anionic dyes are toxic due to the aromatic cycle present in their structure. The degradation of dyes is not easy and can induce the following diseases: dizziness, jaundice, cyanosis, burns, allergic problems, vomiting, diarrhea, nausea, and even effects on development and mental health [4].

Thus, it is essential to remediate waters and wastewater containing dyes using environmentally friendly, cost-effective, and efficient treatment methods. Among all the approaches for the depollution of dye-contaminated waters, comprising chemical methods (coagulation, flocculation, oxidation, etc.), physical (filtration, adsorption, irradiation, ion exchange, membrane processes, etc.), and biological approaches (microbial discoloration), adsorption is the most commonly used process owing to the simplicity and wide variety of existing natural and man-made adsorbent materials.

Numerous types of materials such as activated carbon [5], zeolites [6], lignocellulosic materials [7], clays [8], double-layered hydroxides [9], metal-organic networks (MOFs) [10], magnetic materials [11], metal oxides [12], graphene oxide [13], silica [14], polymeric materials [15,16], carbon nanotubes [17], agricultural wastes [18], chitosan [19], food-processing by-products [20], clays [21], dendritic polymers [22], and hydroxyapatite based materials [23] were applied as dye adsorbents in synthetic aqueous solutions and industrial wastewater.

The disadvantages, such as the high cost of adsorbents and the difficulties of separation after the wastewater treatment process, do not promote their widespread use. The separation of pollutant-loaded adsorbents can be performed by centrifugation, free settling, and membrane filtration. These separation methods involve complicated technical equipment and high operational and maintenance costs. Therefore, they have to be replaced by other efficient methods in water/wastewater treatment applications. One of the most important and economically efficient separation methods is magnetic separation [23,24,25,26,27,28]. This can be applied instead of filtration and centrifugation. Thus, it is necessary to develop and test new magnetic materials to remove dyes from wastewater because they are important in this field having high adsorption capacity and magnetic properties that are important in removing pollutants from water/wastewater. Hence, cobalt ferrite (CoFe_2_O_4_) and cobalt ferrite–chitosan composite (CoFe_2_O_4_-Chit) have been prepared by a simple coprecipitation method. The main advantages of preparing cobalt ferrite–chitosan composite are that the obtained microparticles can be applied in batch and column systems, as compared to cobalt ferrite nanoparticles that are suitable to be used only in batch systems, and they can be easily removed from treated water using an external magnetic field. The optimum experimental conditions of the Congo Red and Methyl Orange removal onto magnetic materials were established in batch systems from single and binary solutions. The isothermal and kinetic study was accomplished to emphasize the Congo Red and Methyl Orange removal mechanisms. Desorption tests have been applied to prove the reusability of the materials tested.

## 2. Materials and Methods

### 2.1. Materials

Fe(NO_3_)_3_·9H_2_O 96%, Co(NO_3_)_2_·6H_2_O, chitosan, and glacial acetic acid (analytical grade, Sigma-Aldrich Chemie GmbH, Steinheim, Germany) were used in the synthesis of magnetic materials. NaOH pellets, HCl 35% suprapure and NH_4_OH 25% (analytical grade) were purchased from Merck KGaA Darmstadt, Germany. Congo Red (CR) 99% (C_32_H_22_N_6_Na_2_O_6_S_2_) and Methyl Orange (MO) (ACS reagent, dye content 85 %, C_14_H_14_N_3_NaO_3_S, Sigma-Aldrich Chemie GmbH, Steinheim, Germany) and ultrapure water were used for preparing the dye solutions. Ethyl alcohol (95%) and acetone (99%) (Sigma-Aldrich Chemie GmbH, Steinheim, Germany) were used as desorbing agents, while ammonium acetate 99% and acetonitrile 99% (Merck KGaA Darmstadt, Germany) as mobile phase for chromatographic separation and detection were used.

### 2.2. Characterization Methods and Instruments

The magnetic materials were characterized by analytical techniques such as Fourier- transform infrared (FTIR) analysis, N_2_ sorption analysis, X-ray diffraction, SEM/TEM, and magnetic measurements. FTIR spectra (4000–400 cm^−1^) were registered on a JASCO FT/IR-4700 spectrometer (Tokyo, Japan) using KBr pellets. The specific surface areas (S_BET_) of the magnetic materials were analyzed by N_2_ adsorption–desorption at −196 °C using a Micromeritics ASAP 2020 automatic adsorption system (Norcross, GA, USA). The Brunauer–Emmett–Teller method on the partial pressure adsorption data (P/Po) in the range 0.05–0.3 was involved in determination of S_BET_ values. XRD structural characterization was performed with a Panalytical X’Pert Pro MPD equipment (Malvern, Worcestershire, UK) in Bragg–Brentano configuration, continuous scanning in the range 5°−85° deg 2θ, with a step of 0.02° and an acquisition time varying from 15 s/step to 45 kV and 40 mA to 30 s/step in order to increase the resolution in the case of the CoFe_2_O_4_–Chit sample. A divergent slit was used on the incident beam, and a nickel filter and a curved graphite monochromator were placed on the diffracted beam, obtaining a monochromatic CuKα radiation of λ = 0.15418 nm. Calculations of the average crystallite size and identification of crystalline phases were performed using Panalytical’s HighScore Plus software and the ICDD (International Center of Diffraction Data) database. SEM images of magnetic materials have been recorded on QUANTA FEG-250 (FEI producer, Brno, Czech Republic) type instrument. For microstructural investigations, the JEOL 2100 Transmission Electron Microscope (TEM) equipped with JEOL Energy Dispersive Spectroscopy (EDS) Detector (Akishima, Tokyo, Japan) has been used. The TEM specimen has been prepared using the standard powder method, using a Cu grid. The magnetic properties were measured at room temperature on Lake Shore’s fully integrated Vibrating Sample Magnetometer system 7404 (VSM) (Westerville, OH, USA).

The performance of magnetic materials in CR and MO removal process has been explored in batch tests performed on GFL 3015 orbital shaker (Burgwedel, Germany) at 150 rpm (rotation per minute). An Agilent 3200 laboratory pH-meter (Agilent Technologies, Shanghai, China) was utilized to evaluate the dye solution’s pH.

The chromatographic separation and detection were performed on an Agilent 1200 series HPLC (Tokyo, Japan) in order to determine the CR and MO concentration in initial solutions and after the removal process. The system has a semipermeable membrane degasser, quaternary pump, self-sampler with variable injection volume (0.1–100 µL), thermostatted column compartment, and a Diode Array Detector (DAD) with the capacity to register simultaneously UV-Vis spectra (190–900 nm) and up to 8 discrete wavelengths. Data acquisition, processing, and reporting has been accomplished with Agilent ChemStation software. The mobile phase was 30% aqueous phase (100 mM ammonium acetate, pH 5) and 70% organic solvent (Acetonitrile) (*v*/*v*), which resulted in a narrow and high symmetry Congo Red peak. All chromatographic runs were carried out on an Acclaim Surfactant Plus column (150 × 3.0 mm, 3.0 µm) from Thermo Scientific. The injection volume was 10 µL and column was kept at 30 °C. The chromatogram run-time was only 12 min (Methyl Orange retention time approx. 6 min and Congo Red retention time approx. 12 min) (Figure 1). Detection of the Congo Red and Methyl Orange compounds was carried out at the optimal wavelengths identified after the maximum absorption in UV-Vis severs: λ = 506 nm (Congo Red) and λ = 428 nm (Methyl Orange). Appendix A show the UV-Vis absorption spectra of Congo Red obtained by High-Performance Liquid Chromatography with Diode-Array Detector (HPLC-DAD) method.

Method calibration curves were obtained using Congo Red and Methyl Orange aqueous solutions of: 0.1, 0.2, 0.5, 1, 2.5, 5, and 10 mg/L (Appendix A).

### 2.3. The Adsorbents Synthesis Protocol

Fe(NO_3_)_3_·9H_2_O 96% and Co(NO_3_)_2_.6H_2_O were used as source of Fe(III) and Co(II) ions, respectively, for cobalt ferrite (CoFe_2_O_4_) synthesis. To an aqueous solution containing Fe(NO_3_)_3_·9H_2_O and Co(NO_3_)_2_.6H_2_O in molar ratio (2:1), 60 mL of NH_4_OH 25% solution was added to precipitate the cobalt ferrite. The obtained mixture was heated at 80 °C, under stirring for 3 h. The resulting precipitate was filtered and washed several times with distilled water.

For the synthesis of cobalt ferrite–chitosan composite (CoFe_2_O_4_-Chit), cobalt ferrite nanoparticles synthesized as described above were added to a chitosan gel prepared by dissolving 0.25 g of chitosan in 25 mL of 10% acetic acid. The cobalt ferrite suspension in chitosan was stirred for 1 h (room temperature and 500 rpm) then dropped with a syringe into 500 mL 30% NaOH. CoFe_2_O_4_-Chit particles were stirred for 14 h (at room temperature and 500 rpm) to complete the chitosan precipitation process on the cobalt ferrite surface. Upon completion of the coating process, CoFe_2_O_4_–Chit particles were washed with ultrapure water and dried in an oven at 60 °C for 3 h.

### 2.4. The Adsorption and Desorption Protocol

To study the influence of operational parameters on the adsorption capacity of CoFe_2_O_4_ and CoFe_2_O_4_–Chit, batch tests were performed in the following experimental conditions:(i)A total of 25 mL dye solution in contact with 0.01 g CoFe_2_O_4_/CoFe_2_O_4_–Chit at room temperature;(ii)A pH range of 2.22–10.8; HCl and NH_4_OH solutions of various concentrations have been used for changing the pH of the dye solution;(iii)Contact time between 5 and 360 min;(iv)Initial dye concentration ranged from 4.98 to 102.81 mg/L;(v)Single CR/MO and binary CR + MO solutions were investigated;(vi)Five recyclability experiments were performed by the use of 25 mL of desorbing agent and 0.01 g magnetic materials loaded with dye, for 4 h contact time at 150 rpm and at room temperature.

After adsorption, the adsorbent was removed from solution using a hand-held neodymium magnet, and the residual concentration of dye was determined.

The amount of dye retained per gram of CoFe_2_O_4_/CoFe_2_O_4_–Chit at equilibrium and at various contact times was estimated using the Equation (1):(1)$$Q_{t\;}=\;\frac{\left(C_{0}\;-{\;C}_{t}\right)\;\times \;V}{m}$$
where *Q_t_*—represents the removal/adsorption capacity which is the amount of CR/MO retained per gram of CoFe_2_O_4_/CoFe_2_O_4_–Chit at various contact times (mg/g);

*C_0_*—the CR/MO initial concentration (mg/L);

*C_t_*—the CR/MO concentration at time (*t*) of contact with adsorbent (or at different pH values) (mg/L);

*V*—the dye solution volume (L);

*m*—the amount of CoFe_2_O_4_/CoFe_2_O_4_-Chit used as adsorbent (g).

The desorption efficiency has been calculated by the use of Equation (2) [29]:(2)$$D\left(\%\right)=\;\left(\frac{Q_{D}}{Q_{e}}\right)\times 100$$
where *Q_D_* is desorption capacity calculated with Equation (3) (mg/g).

*Q_e_* means the adsorption capacity at equilibrium (mg/g):(3)$$Q_{D}=\;\frac{C_{f}}{m}\times V$$
where *C_f_* constitutes final concentration of CR/MO desorbed (in solution) (mg/L);

*V* is the eluent volume (L);

*m* represents the amount of CoFe_2_O_4_/CoFe_2_O_4_–Chit loaded with dye (g).

All the experiments have been performed in triplicate and the maximum experimental error is 5%.

### 2.5. The Mathematical Modeling of Adsorption Process

Understanding the adsorbent–adsorbate relationship, the distribution of adsorbed molecules at equilibrium between the solid and liquid systems through the sorption isotherms is useful for explaining the processes of adsorption of pollutants. In addition, the adsorption capacity of adsorbents can be predicted using mathematical modeling of the adsorption process by means of adsorption isotherms. The most used adsorption isotherm models for this purpose are Langmuir and Freundlich isotherms. These isotherms are expressed in the form of the following mathematical equations:(i)Nonlinear form of the Langmuir isotherm equation [30]:
(4)$$Q_{e}=\frac{Q_{max}K_{L}C_{e}}{1+K_{L}C_{e}}$$
where *C_e_* constitutes CR/MO concentration at equilibrium (in solution) (mg/L);

*K_L_* is the equilibrium constant of the Langmuir model connected with the adsorption energy (L/mg);

*Q_e_* means the adsorption capacity at equilibrium (mg/g);

*Q_max_* is the maximum adsorption capacity (mg/g).
(ii)Nonlinear for the Freundlich isotherm equation [31]:
(5)$$Q_{e}=K_{F}\times C_{e}^{\frac{1}{n}}$$
where K_f_ (mg/g) is adsorption capacity determined from Freundlich equation;

1/*n* represents Freundlich parameter with respect to adsorption intensity;

*C_e_* is CR/MO concentration at equilibrium (in solution) (mg/L).

The kinetics of the CR retention process on prepared magnetic materials have been studied for their possible transformation from laboratory-scale experiments to pilot-scale remediation of industrial wastewater containing dyes. Therefore, the kinetic description of CR/MO remediation by adsorption on magnetic materials was performed, using pseudo-first-order, pseudo-second-order kinetic models, and intraparticle diffusion.

The pseudo-first-order kinetic model is mathematically expressed by the Equation (6):(6)$$Q_{t}=Q_{e}(1-e^{-k_{1}t})$$
where *k*_1_ is pseudo-first-order rate adsorption constant (min^−1^), and *Q_e_*, *Q_t_* are the adsorption capacity at equilibrium and, respectively, the amount of CR/MO retained on the adsorbent at time *t* (mg/g).

For the expression of the pseudo-second-order kinetic model, Equation (7) is used [32]:(7)$$Q_{t}=\frac{Q_{e}^{2}k_{2}t}{1+Q_{e}k_{2}t}$$
where k_2_ represents the rate constant of the pseudo-second-order adsorption process (g/mg∙min);

*Q_e_*, *Q_t_* are the adsorption capacity at equilibrium and, respectively, the amount of

CR/MO retained on the adsorbent at time *t* (mg/g).

Equation (8), presented in 1962 by Weber and Morris [33], is used to express the kinetic model of intraparticle diffusion:(8)$$Q_{t\;}=\;k_{id}t^{0.5}\;+\;C$$
where *k_i_* is intraparticle diffusion rate constant (mg/g·min^0.5^) and C (mg/g) defines intersection that provides indications regarding the thickness of the diffusion layer [33].

## 3. Results and Discussion

### 3.1. Materials Characterization

After synthesis, the prepared magnetic materials were characterized in terms of structure, texture, morphology, and magnetic properties. FTIR spectra of CoFe_2_O_4_ and CoFe_2_O_4_–Chit were recorded and discussed comparatively with the chitosan spectrum (Figure 2).

The FTIR spectrum of CoFe_2_O_4_ reveals the presence of two absorption bands (at 600 and 420 cm^−1^) assigned to stretching and bending vibrations of the M–O bonds in tetrahedral and octahedral sites of spinel ferrites, respectively [34]. The presence of adsorbed water molecules is revealed by a very broad absorption band at 3398 cm^−1^ (O–H stretching vibration mode) and by the band at 1629 cm^−1^ corresponding to bending vibration of water molecules.

The FTIR spectrum of chitosan displays two absorption bands at 3344 and 3284 cm^−1^ assigned to -OH and -NH_2_ stretching vibrations, and two bands at 2915 cm^−1^ due to the asymmetric C-H stretching vibrations and 2880 cm^−1^ corresponding to symmetric C-H stretching vibrations. The peaks at 1650 and 1559 cm^−1^ are related to the C=O stretching vibration of amide and the bending vibration of NH_2_ groups, respectively [35]. The bands located at 1374 and 1024 cm^−1^ are characteristic for C-O stretching vibration of C-OH [36] while that at 1419 cm^−1^ is characteristic for –OH primary alcohol groups [37].

The absorption bands of chitosan are identified in the spectrum of CoFe_2_O_4_–Chit, but they are slightly shifted to lower or higher wavenumbers. These displacements indicate the involvement of the functional groups (-OH and -NH_2_) of chitosan in coordination with the metallic ions of cobalt ferrite. The strong absorption peak at 606 cm^−1^ related to the M-O vibrations of cobalt ferrite confirms the successful embedding of CoFe_2_O_4_ in the chitosan matrix.

The specific surface area (S_BET_) of CoFe_2_O_4_ and CoFe_2_O_4_–Chit particles was determined by N_2_ physisorption at −196 °C using the Brunauer–Emmett–Teller method. The S_BET_ values were 199 m^2^/g for cobalt ferrite and 2 m^2^/g for cobalt ferrite–chitosan composite, respectively. There was a significant decrease in the specific surface area by preparing cobalt ferrite–chitosan composite as expected, taking into account that CoFe_2_O_4_–Chit was obtained as microspheres having a compact surface without porosity, as compared to CoFe_2_O_4_ which was obtained as a nanopowder. Figure 3 shows the nitrogen adsorption/desorption isotherm for CoFe_2_O_4_ that is of type IV according to the IUPAC classification, accompanied by a H2-type hysteresis loop. This type of isotherm and hysteresis is specific for mesoporous materials with a uniform porosity. The pore size distribution (inset of Figure 3) is monomodal and ranges between 2.5 and 6 nm with a peak maximum at ~3.5 nm.

The X-ray diffractograms of the prepared magnetic materials and free chitosan are presented in Figure 4.

In the above diffractograms the families of crystalline planes (220), (311), (400), (511), (440) were identified corresponding to the cobalt ferrite CoFe_2_O_4_, (JCPDS no. 01-080-6487) metal oxide with a spinel-type structure that crystallizes in the cubic system, having the network constants a = b = c = 8.381 [Å] and cell Vol = 588.69 [Å^3^]. In the case of our CoFe_2_O_4_ sample, the network constants a = b = c = 8.344 (6) and Vol = 580.84 and an average crystallite size were calculated with the Debye–Scherrer formula of approximately 4 nm (the average crystallite size was calculated using the Full with at Half Maximum (FWHM) value corresponding to three diffraction maxima, after drawing a baseline and approximating the graph with the pseudo-Voigt function).

In the case of the X-ray diffractogram of chitosan, the crystalline planes (002) and (110) were identified at an angle of 2θ corresponding to the values of 9.68° and 20.05°; the average crystallite size is ≈3nm. These observations are in line with the literature data [38].

In the case of cobalt ferrite–chitosan composite, a significant decrease in the maximum intensity and an increase in the maximum diffraction width can be observed due to the phenomena induced by the very small particle size (a few nanometers) and the stress in the crystal lattice. In the case of CoFe_2_O_4_, a decrease in the network constant “a” and the volume of the elementary cell was observed. However, both crystallographic phases (CoFe_2_O_4_ and chitosan) are present and clearly distinguished.

TEM imaging of CoFe_2_O_4_ (Figure 5) revealed agglomerated nanoparticles (NPs) of ~3 nm in size, almost regular and spherical in shape. TEM observations also confirmed the porous structure of CoFe_2_O_4_. The Selected Area Electron Diffraction (SAED) correlated with the EDS results (Fe:Co ratio of 2:1) proved the CoFe_2_O_4_ phase of the NPs.

SEM images of CoFe_2_O_4_ (Figure 6) showed aggregates of nanoparticles with dimensions between 141 and 432 nm, but also larger aggregates (1.853 μm/2.428 μm/292.8 nm). The tendency of agglomeration of CoFe_2_O_4_ nanoparticles, attributed mainly to the higher interactions between nanoparticles, determined the formation of highly porous aggregates with higher adsorption capacity.

Higher hemispherical particles with diameters between 804 μm and 1.41 mm have been observed for CoFe_2_O_4_–Chit (Figure 7). The shape and surface morphology of CoFe_2_O_4_–Chit particles is different compared to CoFe_2_O_4_. Thus, CoFe_2_O_4_–Chit is characterized by well-defined and smoother particles. Based on these observations it can be remarked that chitosan determined the decrease in the agglomeration tendency characteristic for CoFe_2_O_4_ nanoparticles.

The magnetic properties of CoFe_2_O_4_ and CoFe_2_O_4_–Chit were measured on a VSM at room temperature by recording the magnetization versus applied field curves (Figure 8). Data presented in Figure 8a,b show no room temperature remanence or coercivity, thus revealing a superparamagnetic behavior of both samples. The experimental data were analyzed by fitting to the Langevin function. The saturation magnetization (Ms) values are 33.6 emu/g for CoFe_2_O_4_ and 8.4 emu/g for CoFe_2_O_4_–Chit. The decrease in Ms value of CoFe_2_O_4_–Chit compared to CoFe_2_O_4_ is due to the presence of the diamagnetic chitosan molecules.

### 3.2. Adsorption Studies

The process of removing CR/MO by adsorption on CoFe_2_O_4_ and CoFe_2_O_4_–Chit was studied in batch experiments. The effect of some significant parameters on the retention capacity of these materials was considered. Among the parameters investigated, the following can be mentioned: pH, contact time, and concentration of CR/MO in the initial solution.

#### 3.2.1. The pH Effect on Adsorption Capacity

pH is one of the great significant factors that influence the adsorption process of pollutants from aqueous effluents and wastewater. The pH influences the dyes’ solubility in water, the activity of the functional sites on the surface of the adsorbents, and competition of ions for the adsorption centers.

To study the variation of CoFe_2_O_4_ and CoFe_2_O_4_–Chit adsorption capacity versus pH values, batch experiments were performed in the pH range of 2.22–10.8 using HCl and NH_4_OH solutions of various concentrations. A CR single solution of 102.81 mg/L (experimentally determined) and an MO single solution of 100 mg/L were used in the tests. The binary solutions CR/MO (100 mg/L CR + 100 mg/L MO) with a volume of 25 mL and 0.01 g of adsorbents were used to establish the pH effect on the adsorption capacity. The room temperature experiments were performed for 6 h to reach equilibrium. The experiments were carried out using single and binary dyes solutions at the same pH range. The experimental results regarding the effect of pH on CR/MO adsorption from single solutions are presented in Figure 9.

Figure 9 reveals that in the case of CR adsorption, the studied magnetic materials are characterized by different behaviors. The adsorption capacity of CoFe_2_O_4_ decreases with the increase in dye solution pH from 2 to 11. This can be attributed to the fact that at low pH values, the positively charged CoFe_2_O_4_ surface interacts strongly with the negative sulfonate groups of the dyes. As the pH increases, the adsorption capacity starts to decrease because the surface of the adsorbent becomes less positive, while the molecules of adsorbate pass in the anionic form. The electrostatic repulsion forces become stronger in basic medium, thereby lowering the adsorption. The highest adsorption capacity of CoFe_2_O_4_ was noticed at pH around 4.5. The adsorption behavior of CoFe_2_O_4_–Chit is slightly different. In an acidic pH range, the high concentration of protons determines the protonation of amine (-NH_2_) and hydroxyl groups (-OH) of chitosan. These protonated functional groups will be electrostatically attracted by the negatively charged sulfonate groups of CR. Repulsive forces with protonated functional groups of Congo Red can also occur at low pH values (Figure 10).

At higher pH values ranging from slightly acidic to neutral, where the chitosan surface is neutral the adsorption of the dye could also occur only by physical forces (hydrogen bonding and van der Waals forces) that will affect the adsorption capacity [39]. The highest value of adsorption capacity has been recorded at pH equal to 10.8. The literature data present similar behaviors related to the sorption properties of chitosan coated magnetic iron oxides [40]. Figure 9 shows that CoFe_2_O_4_ has a higher CR adsorption capacity than CoFe_2_O_4_–Chit. This is related mainly to the higher specific surface area of CoFe_2_O_4_ compared to CoFe_2_O_4_–Chit. On the other hand, the limited adsorption capacity of CoFe_2_O_4_–Chit could be attributed to the fact that during the synthesis of the composite, some functional groups of chitosan react with those of the CoFe_2_O_4_, the number of active sites available to interact with the dye molecules being reduced [41].

Taking into account these results, the further tests regarding adsorption capacity versus time for CR adsorption process were carried out at two pH values (4.5 and 10.8).

By analyzing Figure 9, it can be observed that the maximum adsorption capacity for MO of both adsorbents is at pH = 2.22 in single solutions. With the increase in pH values to 10.8, the adsorption capacity of CoFe_2_O_4_ and CoFe_2_O_4_-Chit decreases sharply. These results indicate that optimum pH value for the adsorption of MO onto CoFe_2_O_4_ and CoFe_2_O_4_–Chit in single solutions is 2.22. This can be attributed to strong electrostatic forces between anionic MO (pKa of MO 4.4–5.5) [42] and the positively charged CoFe_2_O_4_ surface under acidic conditions. The higher value of CoFe_2_O_4_–Chit under acidic conditions is based by the interactions between protonated amino groups of chitosan and sulfonate groups of MO [43]. At higher pH values, the high concentration of negative charges on the surface of both adsorbents inhibits the adsorption of the anionic MO.

Furthermore, it was also observed that both magnetic materials have lower adsorption capacity for MO compared to CR in single solutions, suggesting a higher selectivity of CoFe_2_O_4_ and CoFe_2_O_4_–Chit for CR in these conditions.

The effect of pH on MO/CR adsorption process in binary solution is presented in Figure 11.

A similar behavior has been observed for the pH effect on adsorption capacity of dyes from binary solutions. For both magnetic materials, a decrease in the dyes’ adsorption capacity was observed with the increase in pH value. A decrease in adsorption capacity was recorded in the experiments performed with binary dyes solutions. In case of CR, the maximum experimental adsorption capacity decreases with 35.63 mg/g for adsorption onto CoFe_2_O_4_ and with 49.75 mg/g for adsorption onto CoFe_2_O_4_–Chit from single to binary solutions. A lower decrease in the maximum experimental adsorption capacity was determined for MO (6.5 mg/g for adsorption onto CoFe_2_O_4_ and 39 mg/g for adsorption onto CoFe_2_O_4_–Chit from single to binary solutions). Therefore, at the same concentration, in binary solutions, MO has a more inhibitory effect on CR adsorption onto magnetic materials than CR on MO adsorption.

#### 3.2.2. The Contact Time Effect on Adsorption Capacity

Contact time is a crucial operational parameter in the process of removing pollutants from wastewater and synthetic aqueous solutions. Batch experiments at room temperature and 150 rpm and different time periods (between 5 and 360 min) with volumes of 25 mL of 102.81 mg CR/L and 0.01 g CoFe_2_O_4_/CoFe_2_O_4_–Chit were performed to establish the equilibrium time.

The same experimental conditions were applied in tests for establishing the effect of contact time onto CoFe_2_O_4_/CoFe_2_O_4_–Chit adsorption capacity of MO and CR/MO from binary solutions.

The variation of the amount of dye retained per gram of CoFe_2_O_4_/CoFe_2_O_4_–Chit versus contact time is illustrated in Figure 12, Figure 13 and Figure 14.

It can be stated that the process of CR and MO retaining onto CoFe_2_O_4_/CoFe_2_O_4_–Chit takes place in two stages. In the first phase occurring in the first 30 min, the amount of retained CR/MO rapidly increases due to the fact that during this period the surface of the adsorbents has a large number of active centers available for retaining CR/MO (pollutant). The same finding is valid for the adsorption process of CR and MO from binary solutions.

As the contact time increases, the number of free active centers decreases, and consequently the speed of the retention process decreases. In our case, the equilibrium time is approximately 180 min for CR adsorption and 240 min for MO adsorption from single and binary solutions.

It can also be seen from the Figure 12 that the value of the maximum CR adsorption capacity experimentally determined is 136.13 mg/g for CoFe_2_O_4_ and 75.25 mg/g for CoFe_2_O_4_–Chit. Figure 13 reveals that the value of the maximum MO adsorption capacity experimentally determined is 92.2 mg/g for CoFe_2_O_4_ and 64.5 mg/g for CoFe_2_O_4_–Chit. These values are lower for binary solutions (Figure 14). Thus, the maximum CR adsorption capacity experimentally determined decreases to 100.5 mg/g for CoFe_2_O_4_ and 25.5 mg/g for CoFe_2_O_4_-Chit, while the maximum MO adsorption capacity experimentally determined decreases to 85.7 mg/g for CoFe_2_O_4_ and 54.5 mg/g for CoFe_2_O_4_–Chit, respectively.

These values are consistent with the size of the adsorbent particles and the specific surface areas. Cobalt ferrite was obtained as a nanopowder with specific surface area of about 199 m^2^/g, while CoFe_2_O_4_–Chit ferrite was obtained as microparticles with a specific surface area of about 2 m^2^/g.

#### 3.2.3. The Isothermal Study

##### Single Component Adsorption Isotherm Study

The experimental results of the dyes’ adsorption process on CoFe_2_O_4_/CoFe_2_O_4_–Chit from single-component solutions were fitted with Langmuir and Freundlich isotherm models (Figure 15 and Figure 16).

The Langmuir isotherm model assumes a monolayer adsorption on a homogeneous surface. Consequently, all the adsorption centers are identical [32].

The Freundlich isotherm describes a multilayer adsorption on a heterogeneous and non-uniform surface [32].

The equilibrium isotherm parameters and the correlation coefficients (R^2^) for the studied adsorption processes are presented in Table 1.

For the Langmuir isotherm, a dimensionless separation factor (R_L_) can be estimated with Equation (9):(9)$$R_{L}=\frac{1}{\left(1+K_{L}\times C_{0}\right)}$$
where C_0_ is the highest initial concentration of dye (mg/L);

K_L_ is adsorption constant for Langmuir model (L/mg).

The R_L_ value can be used to determine whether the Langmuir model favorably describes the adsorption process. A value of R_L_ > 1 indicates that the adsorption is not favorable; if R_L_ = 1, the adsorption is linear; for 0 < R_L_ < 1, the adsorption is favorable, and if R_L_ = 0, it means that adsorption is irreversible [33].

The goodness of fit of the experimental data has been verified by the calculation of the correlation coefficient (R^2^) and the Akaike’s Information Criterion (AIC). Lower AIC values (on a scale from −∞ to +∞) demonstrate that the respective model is more likely to characterize the sorption process than the alternative model [44].

As can be seen in Table 1, the correlation coefficients (R^2^) and AIC values reveal that all the adsorption data fit better with the Langmuir isotherm model. These results suggest that adsorption of both dyes on the studied magnetic materials from single-component solutions takes place as a monolayer adsorption process on a homogeneous surface. The R_L_ values, between 0 and 1, support the favorability of the adsorption reaction and confirm that electrostatic interactions between magnetic adsorbents and dye molecules.

##### Competitive Adsorption of CR and MO in Binary Solutions

For a binary mixture containing CR and MO, the adsorption capacity of CR can be determined using the modified Langmuir isotherm model, mathematically expressed by the Equation (10) [45]:(10)$$Q_{e,CR}=\;\frac{Q_{max,CR}K_{L,CR}C_{e,CR}}{1+K_{L,CR}C_{e,CR}+K_{L,MO}C_{e,MO}}$$

The linearization form of this model is given by Equation (11):(11)$$\frac{1}{Q_{e,CR}}=\;\frac{1}{Q_{max,CR}}+\frac{1}{Q_{max,CR}K_{L,CR}}\;\left[\frac{1}{C_{e,CR}}+\;\frac{K_{L,MO}C_{e,MO}}{C_{e,CR}}\right]\;$$

To estimate MO adsorption capacity in the binary mixture, Equation (9) was rewritten as shown in Equation (12).
(12)$$\frac{1}{Q_{e,MO}}=\;\frac{1}{Q_{max,MO}}+\frac{1}{Q_{max,MO}K_{L,MO}}\;\left[\frac{1}{C_{e,MO}}+\;\frac{K_{L,CR}C_{e,CR}}{C_{e,MO}}\right]\;$$
where *C_e,CR_*, *C_e,MO_*, *Q_e,CR_*, and *Q_e,MO_* are equilibrium concentration and equilibrium adsorption capacity of *CR* and *MO* in the binary solution, *K_L,CR_* and *K_L,MO_* are Langmuir constants obtained in singe dye systems, *Q_max,CR_* and *Q_max,MO_* are maximum adsorption capacity of CR and MO in the binary solution.

Substituting the values of *C_e,CR_*, *C_e,MO_*, *Q_e,CR_*, *K_L,CR_*, and *K_L,MO_* in Equation (11) and making a linear plot of $\frac{1}{Q_{e,CR}}$ versus $\left[\frac{1}{C_{e,CR}}+\;\frac{K_{L,MO}C_{e,MO}}{C_{e,CR}}\right]$ (Appendix A), one can obtain *Q_max,CR_* in the binary solution. *Q_max,MO_* can be calculated by the same procedure using the Equation (12).

According to Mahamadi et al. [46], the ratio between Q_max,binary_ and Q_max,single_ gives information regarding the adsorption dynamics in binary solutions. When this ratio is >1, the two adsorbates have a synergistic behavior, the effect of the mixture being greater than the effect of the individual adsorbates in the mixture. When the ratio is <1, the two adsorbates have an antagonistic behavior, the effect of the mixture being less than that of each of the individual adsorbates in the mixture. When the ratio = 1, the mixture has no effect on the adsorption of each of the adsorbates in the mixture. The Q_max,binary_/Q_max,single_ ratios for the adsorption of CR and MO onto CoFe_2_O_4_ and CoFe_2_O_4_–Chit are presented in Table 2.

The Q_max,binary_/Q_max,single_ ratio is <1 only in the case of CR adsorption onto CoFe_2_O_4_, suggesting that the adsorption of this dye onto CoFe_2_O_4_ is hindered by the presence of MO. The effect of the mixture seems to be synergistic in the case of adsorption of both dyes onto CoFe_2_O_4_–Chit and for MO onto CoFe_2_O_4_. This behavior can be attributed to the size of the dye molecules and the texture of the adsorbent. As the CR molecules are much larger than those of MO, it is expected that their adsorption on CoFe_2_O_4_ with a porous structure (pore diameters ranging from 2.5 to 6 nm) will be hindered by the presence of MO molecules in a binary solution. In the case of MO adsorption on CoFe_2_O_4_ from a binary solution, the CR molecules, being larger, cannot penetrate as easily into the porous network of the adsorbent; therefore, they do not depress the MO adsorption. On the contrary, they have a synergistic effect. In the case of CoFe_2_O_4_–Chit which is not porous, the adsorption of the dyes is not influenced by the mutual presence of both components in solution because the process occurs on the external surface of the adsorbent.

Comparing the maximum adsorption capacities of both magnetic adsorbents included in this study with those reported in the literature for other magnetic materials, one can notice that they present comparable or even higher values (Table 3).

#### 3.2.4. The Kinetic Study

Pseudo-first-order, pseudo-second-order, and the intraparticle diffusion kinetic models were applied to determine the controlling mechanism of dye adsorptions from aqueous solution. Graphical representations of these kinetic models are displayed in Appendix A, and the calculated kinetic parameters and values of regression coefficients are presented in Table 4.

Analyzing the results, it can be noticed that in the case of single-component solutions, the correlation coefficients R^2^ for the pseudo-first-order and pseudo-second-order kinetic models have close values regardless of adsorbate and adsorbent. Therefore, using a non-linear method, both pseudo-first-order and pseudo-second-order kinetics represent well the kinetics of the adsorption process. Taking into account the AIC parameter and the fact that calculated Q_e_ values are much closer to those determined experimentally in the case of pseudo-first-order kinetic model, it can be said that this model describes somewhat better the adsorption process in single-component solutions.

In binary solutions it can be noticed that adsorption of CR/MO on CoFe_2_O_4_ follows better the pseudo-second-order kinetics model as the R^2^ coefficients are higher and the AIC values are smaller than those calculated for pseudo-first-order model. The adsorption process of CR/MO on CoFe_2_O_4_–Chit seems to respect the kinetic behavior identified for single-component solutions.

Furthermore, it can be observed (from Appendix A) that for MO adsorption and CR/MO adsorption from binary solutions, the curve that describes the intraparticle diffusion kinetic model can be fragmented into three lines with individual slopes, but the lines do not pass through the origin. According to this model, if the plot is multilinear or does not pass through the origin, the process of adsorption is controlled by two or more diffusion mechanisms with different rate constants [58]. The literature data indicate that the first section of the curve can be attributed to the bulk diffusion; the linear section is characteristic to the intraparticle diffusion and the plateau to the equilibrium [59]. This means that in our case the adsorption process occurs in phases, and pore diffusion may not be considered as the sole operating rate-controlling step.

#### 3.2.5. Desorption Study

One of the most important properties of any adsorbent is related to its regeneration capacity and possibility to be used in multi-cycle systems. These properties influence the performance of the adsorbents and the total operational cost of the adsorption process. In our case, acetone and ethanol were designated as desorbing agents in order to test the reusability of CoFe_2_O_4_ and CoFe_2_O_4_–Chit in multiple dye adsorption–desorption cycles. The reason of their choice was their high dipole moments [60]. Five adsorption–desorption cycles in acetone and ethanol were applied to check the reusability of both adsorbents. The results of these experiments are presented in Figure 17a,b.

As shown in Figure 17a,b, the adsorption capacity was maintained at the same level after three cycles of adsorption–desorption for both desorbing agents tested. A slight decrease in the adsorption capacity was observed after five consecutive runs: the adsorption capacity of CoFe_2_O_4_ decreased by 12.75% for CR and 7.35% for MO by the use of acetone as a desorbing agent. A higher decrease has been recorded by using ethanol as a desorbing agent (17.47% for CR and 9.50% for MO). A similar conclusion can be drawn from the experiments performed for CoFe_2_O_4_–Chit. In the case of CR, we observed a decrease by 21.81% and 18.73% by the use of acetone and ethanol, respectively. A lower decrease in the adsorption capacity of CoFe_2_O_4_–Chit was noticed after five MO adsorption–desorption cycles: 8.88% by the use of acetone and 10.60% by the use of ethanol. These values revealed that after five adsorption–desorption cycles, the adsorption capacities are still at high level.

Figure 18 shows the desorption efficiency behavior for all five cycles of adsorption-desorption.

The results showed that both desorbing agents have similar desorption efficiency. The desorption efficiency of acetone varied between 99.5 and 80.33%, while the desorption efficiency of ethanol varied between 99.44 and 81.27%. After five adsorption–desorption consecutive cycles, both adsorbents lost 18–19% of the desorption efficiency.

Thus, it can be said that both magnetic materials can be effective adsorbent materials for the remediation of dye-polluted waters.

## 4. Conclusions

Cobalt ferrite and cobalt ferrite–chitosan composite were synthesized by a simple coprecipitation method to be used as effective adsorbents for removal of Congo Red and Methyl Orange dyes from single and binary aqueous solutions. The as-obtained magnetic materials prepared were characterized by FTIR spectroscopy, specific surface area and porosity analysis, SEM and TEM analysis, X-Ray diffraction, and magnetic measurements. The results confirmed the successful synthesis of magnetic materials.

The optimal parameters of the CR/MO retention process on the prepared cobalt ferrite/cobalt ferrite–chitosan composite were determined from batch experiments at room temperature in single and binary dye solutions. It was found that the adsorption process depends on pH, time, initial concentration of CR/MO, and the presence of competing dye. The amount of CR/MO retained per gram of adsorbent material increases with increasing initial CR/MO concentration and contact time until equilibrium is reached. Experimental data were correlated with the most commonly used models of adsorption isotherms to describe the adsorption processes. The Langmuir isotherm best describes the adsorption on both adsorbents and indicates that CR/MO adsorption occurs on the homogeneous surface of cobalt ferrite and cobalt ferrite–chitosan composite as a monolayer. The values of adsorption capacity are higher for cobalt ferrite than cobalt ferrite–chitosan composite. These values are consistent with the size and specific surface areas of the adsorbent particles. Additionally, the values of maximum adsorption capacity for cobalt ferrite revealed that this adsorbent is selective for CR, while cobalt ferrite–chitosan composite is selective for MO.

Kinetic modeling, by the non-linear method, demonstrates that the kinetics of the adsorption process of CR/MO on cobalt ferrite/cobalt ferrite–chitosan composite are different in single-component and binary solutions. Pore diffusion might have significant influence on the kinetics of the process. Acetone and ethanol can be successfully used as desorbing agents, the adsorption capacities of the magnetic materials having a small decline after five adsorption–desorption cycles.

The results of the study indicate that cobalt ferrite and cobalt ferrite–chitosan composite are relevant as magnetic adsorbents to clean dye-containing wastewater.

## Figures and Tables

**Figure 1 nanomaterials-11-00711-f001:**
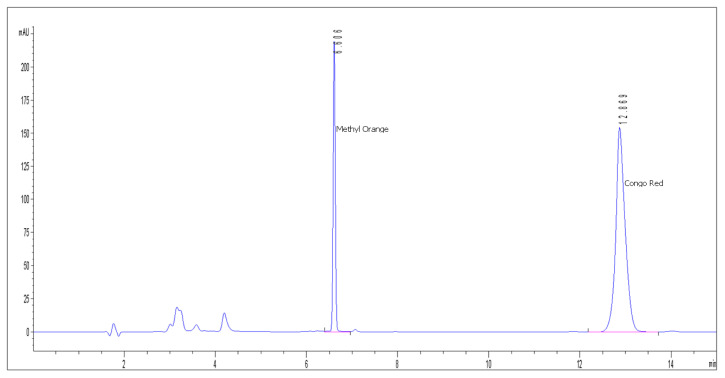
Chromatogram obtained from the analysis of Congo Red (CR) and Methyl Orange (MO).

**Figure 2 nanomaterials-11-00711-f002:**
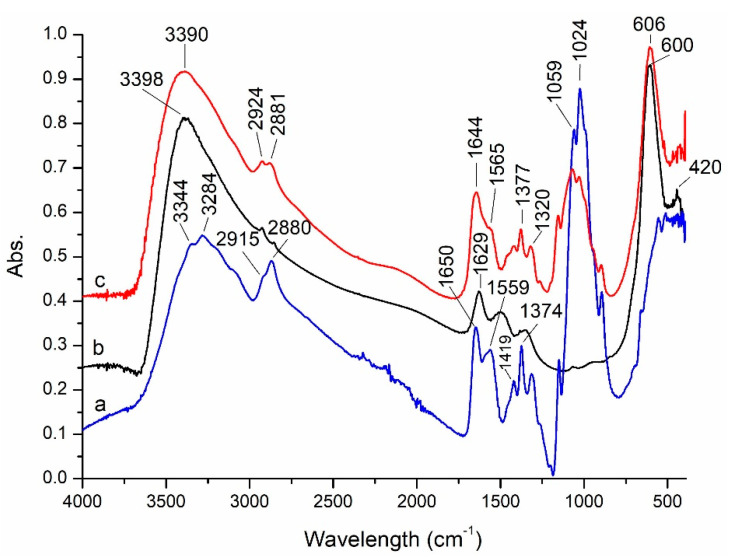
FTIR spectra of: (a) chitosan; (b) CoFe_2_O_4_; (c) CoFe_2_O_4_−Chit.

**Figure 3 nanomaterials-11-00711-f003:**
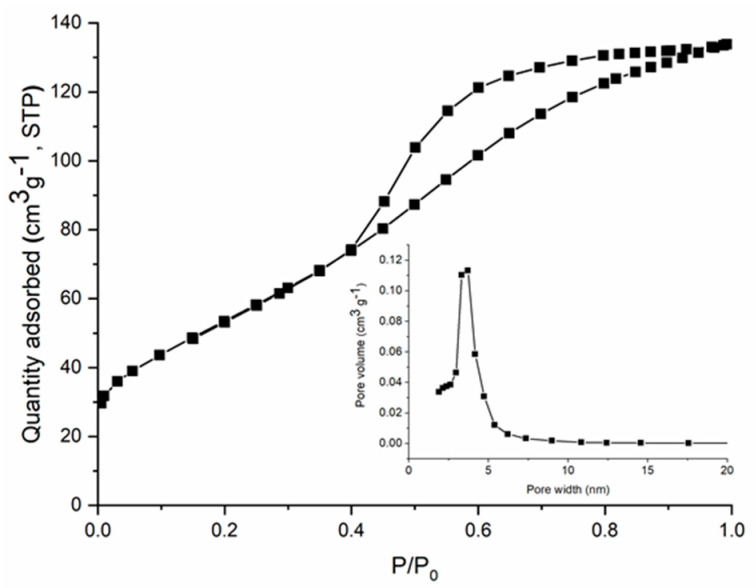
N_2_ adsorption−desorption isotherms and pore size distribution (inset) of CoFe_2_O_4._

**Figure 4 nanomaterials-11-00711-f004:**
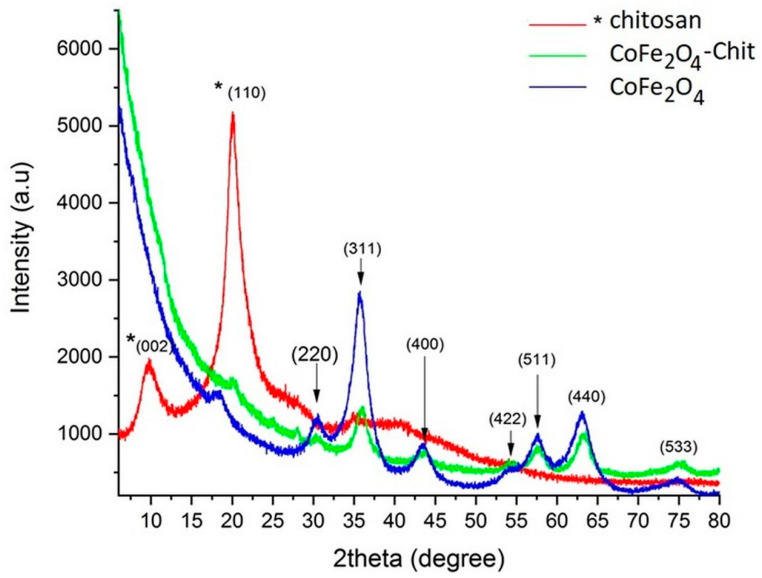
X-ray diffractograms of CoFe_2_O_4_, CoFe_2_O_4_–Chit, and Chitosan.

**Figure 5 nanomaterials-11-00711-f005:**
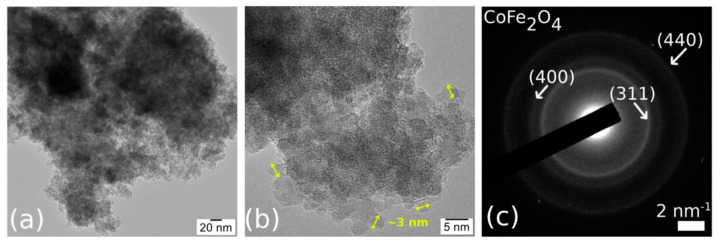
TEM image (**a**), High-Resolution Transmission Electron Microscopy (HRTEM) image (**b**), and Selected Area Electron Diffraction (SAED) (**c**) of the CoFe_2_O_4_ sample.

**Figure 6 nanomaterials-11-00711-f006:**
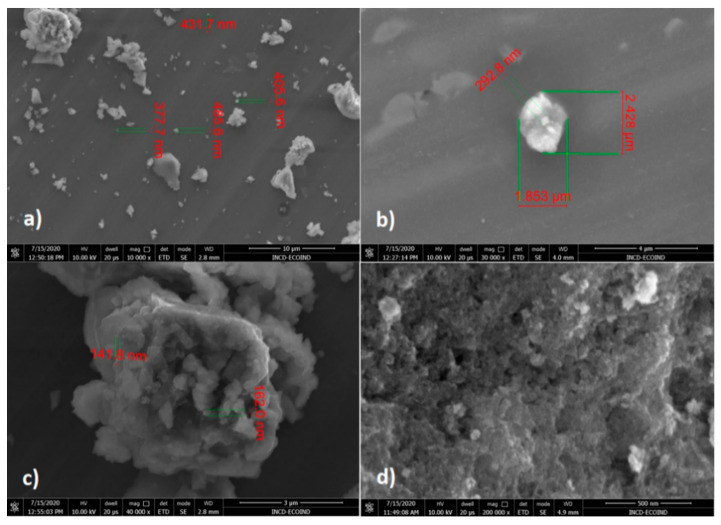
SEM images of CoFe_2_O_4_ at different magnifications: (**a**) 10.000×, (**b**) 30.000×, (**c**) 40.000×, (**d**) 200.000×.

**Figure 7 nanomaterials-11-00711-f007:**
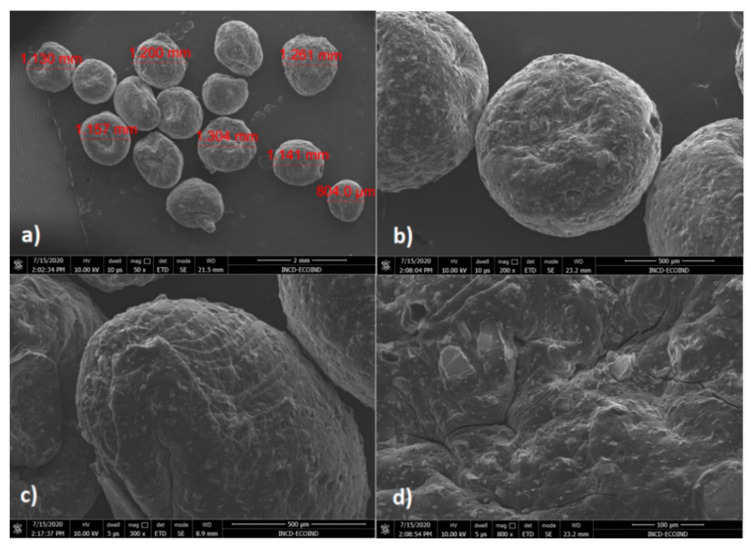
SEM images of CoFe_2_O_4_–Chit at different magnifications: (**a**) 50×, (**b**) 200×, (**c**) 300×, (**d**) 800×.

**Figure 8 nanomaterials-11-00711-f008:**
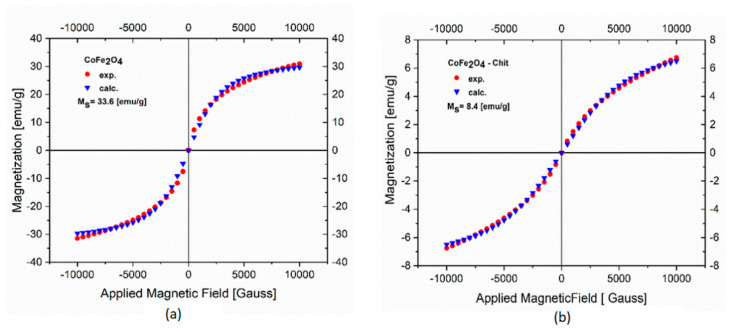
Magnetic hysteresis curves of CoFe_2_O_4_ (**a**) and CoFe_2_O_4_−Chit (**b**) at room temperature.

**Figure 9 nanomaterials-11-00711-f009:**
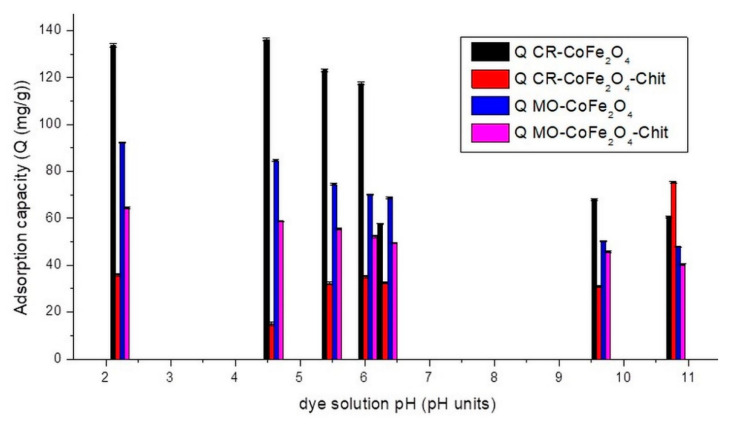
The variation of adsorption capacity versus the CR/MO solution’s pH (single system).

**Figure 10 nanomaterials-11-00711-f010:**
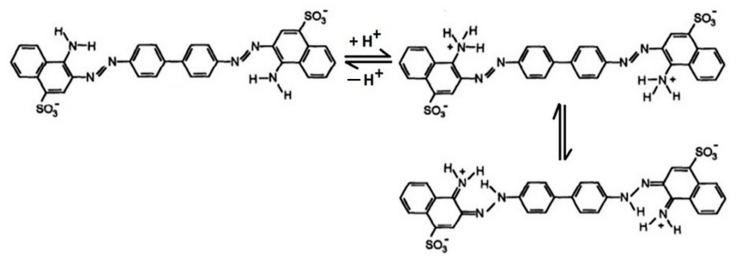
The Congo Red structures in acidic solution [39].

**Figure 11 nanomaterials-11-00711-f011:**
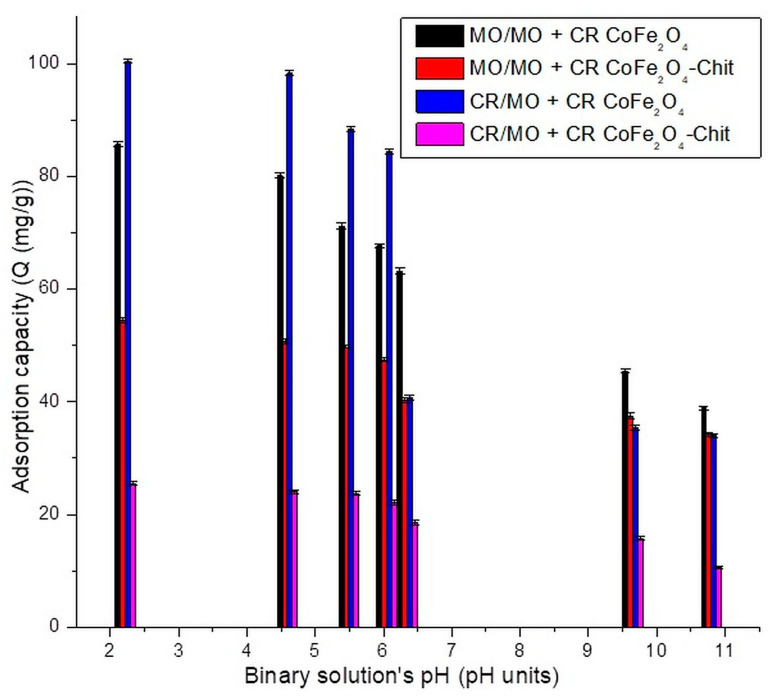
The variation of adsorption capacity versus CR/MO solution pH (binary system).

**Figure 12 nanomaterials-11-00711-f012:**
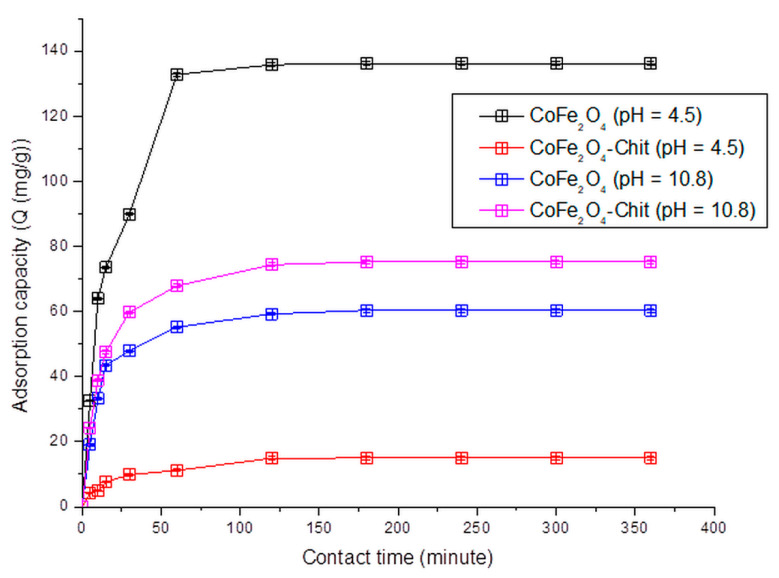
The variation of CoFe_2_O_4_/CoFe_2_O_4_–Chit adsorption capacity versus contact time at pH = 4.5 and pH = 10.8 for the CR adsorption process.

**Figure 13 nanomaterials-11-00711-f013:**
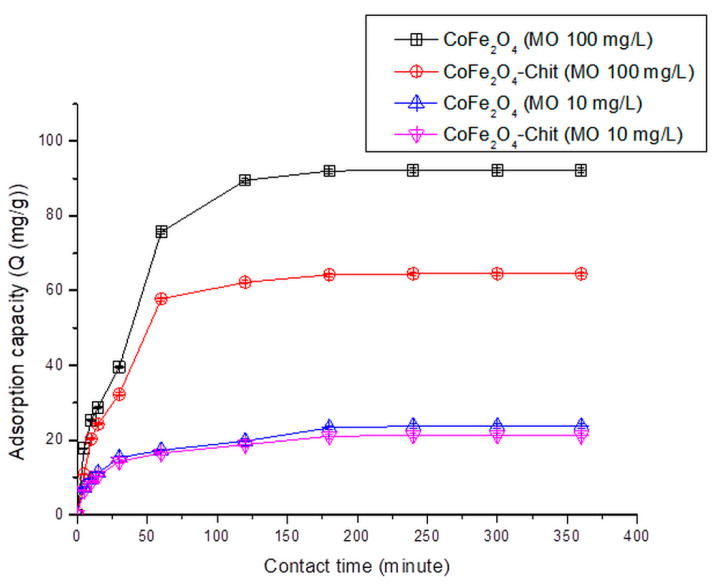
The variation of CoFe_2_O_4_/CoFe_2_O_4_–Chit adsorption capacity versus contact time at pH = 2.22 for the MO adsorption process (pH = 2.22).

**Figure 14 nanomaterials-11-00711-f014:**
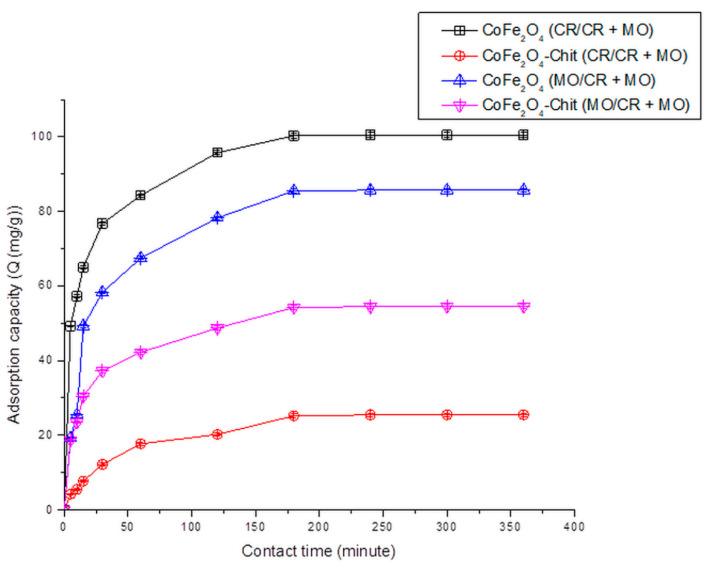
The variation of CoFe_2_O_4_/CoFe_2_O_4_–Chit adsorption capacity versus contact time for the CR and MO adsorption from binary solution (100mg/L CR + 100 mg/L MO) (pH = 2.22).

**Figure 15 nanomaterials-11-00711-f015:**
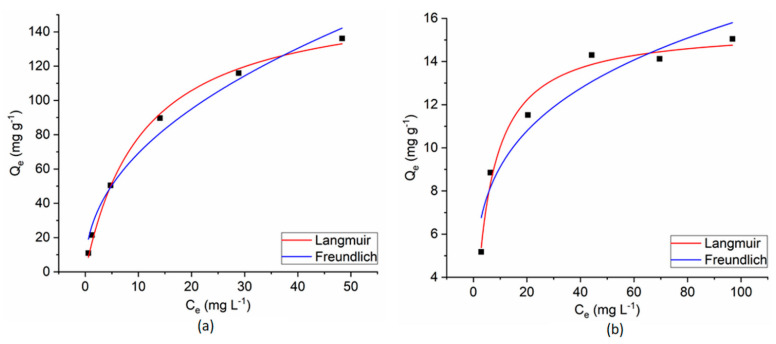
Langmuir and Freundlich fitting curves for CR adsorption onto CoFe_2_O_4_ (**a**) and CoFe_2_O_4_−Chit (**b**) from a single component solution.

**Figure 16 nanomaterials-11-00711-f016:**
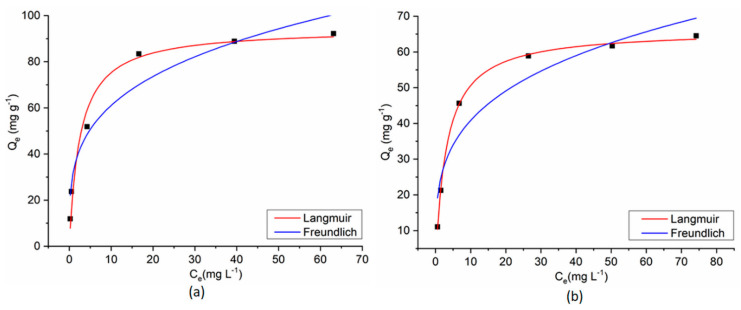
Langmuir and Freundlich fitting curves for MO adsorption onto CoFe_2_O_4_ (**a**) and CoFe_2_O_4_−Chit (**b**) from single component solution.

**Figure 17 nanomaterials-11-00711-f017:**
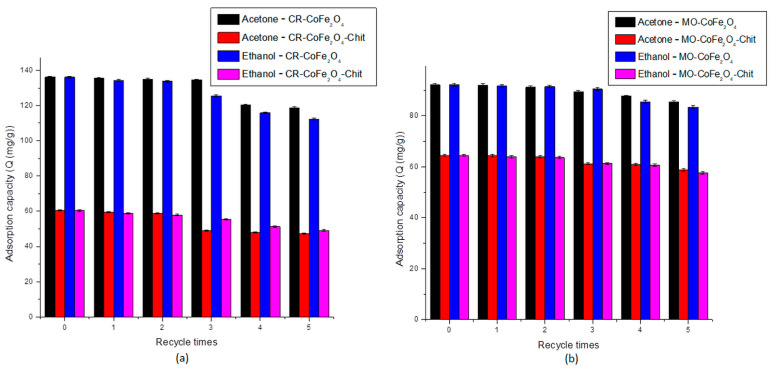
Reusability of CoFe_2_O_4_ and CoFe_2_O_4_−Chit for CR (**a**) and MO (**b**).

**Figure 18 nanomaterials-11-00711-f018:**
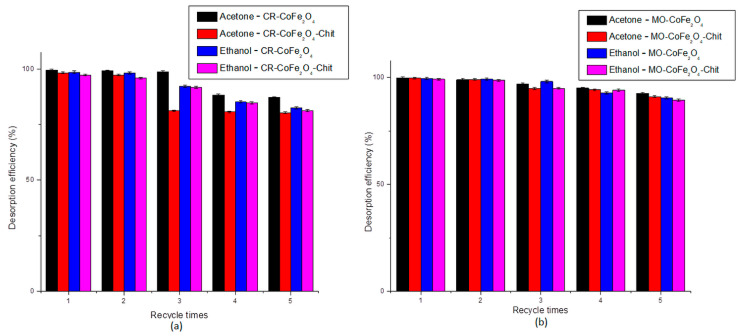
Desorption efficiency of acetone and ethanol for CR (**a**) and MO (**b**).

**Table 1 nanomaterials-11-00711-t001:** Langmuir and Freundlich parameters for the dyes’ adsorption onto CoFe_2_O_4_ and CoFe_2_O_4_–Chit from single-component solutions.

Dye	CR		MO	
Sample	CoFe_2_O_4_	CoFe_2_O_4_-Chit	CoFe_2_O_4_	CoFe_2_O_4_–Chit
Langmuir Parameters				
Q_max_ (mg/g)	162.676 ± 6.0642	15.6042 ± 0.4257	94.4626 ± 4.6652	66.1876 ± 0.5552
K_L_ (L/mg)	0.0928 ± 0.0104	0.1800 ± 0.0233	0.3921 ± 0.1071	0.3251 ± 0.0138
R^2^	0.9964	0.9834	0.9711	0.9988
AIC	30.37	8.51	36.93	12.23
R_L_	0.100	0.050	0.025	0.030
Freundlich Parameters				
K_F_ (mg/g)	24.1787 ± 3.6051	5.2154 ± 0.8263	32.8071 ± 5.5190	22.2068 ± 4.3014
1/n	0.4568 ± 0.0437	0.2424 ± 0.0410	0.2698 ± 0.0480	0.2647 ± 0.0530
R^2^	0.9830	0.9226	0.9236	0.8939
AIC	39.68	17.75	42.77	39.52

**Table 2 nanomaterials-11-00711-t002:** Maximum adsorption capacity calculated using Langmuir and modified Langmuir isotherm models for single-component and binary solutions.

Adsorbent	Dye	Parameters	Single Component Solution (mg/g)	Binary Solution (mg/g)	Q_max,binary/_Q_max,single_
CoFe_2_O_4_	CR	Q_max,CR_	162.67	79.87	0.49
	MO	Q_max,MO_	94.46	117.50	1.24
CoFe_2_O_4_-Chit	CR	Q_max,CR_	15.60	25.32	1.62
	MO	Q_max,MO_	66.18	81.30	1.22

**Table 3 nanomaterials-11-00711-t003:** Adsorption capacities of different magnetic adsorbents from the literature for the removal of CR and MO.

Dye	Adsorbent	Adsorption Capacity (mg/g)	Reference
CR	Fe_3_O_4_@SiO_2_@ZnTDPAT	17.73	[47]
	m-Cell/Fe_3_O_4_/ACCS	66.1	[48]
	MgFe_2_O_4_-NH_2_ NPs	71.4	[49]
	Fe_3_O_4_@SiO_2_@MgAl-borate LDH	158.98	[50]
	Fe_x_Co_3-x_O_4_	128.6	[51]
	Chitosan/iron oxide nanocomposite films	25.5	[40]
	Chitosan/iron oxide nanocomposite films prepared by sonication	700	[40]
	CoFe_2_O_4_	162.68	This study
	CoFe_2_O_4_–Chit	15.60	This study
	CoFe_2_O_4_ from binary solutions with MO	79.87	This study
	CoFe_2_O_4_-Chit from binary solutions with MO	25.32	This study
MO	Muscovite supported Fe_3_O_4_ nanoparticles	149.25	[52]
	Multi-walled carbon nanotubes (MWCNTs) coated with magnetic ZnLa_0.02_Fe_1.98_O_4_ clusters	81	[53]
	Mesoporous Fe_3_O_4_–SiO_2_–TiO_2_ (MFST)	2.5	[54]
	Magnetic iron oxide/carbon nanocomposites	72.68	[55]
	Magnetic iron oxide/carbon nanocomposites from binary solutions with phenol	71.02	[56]
	Rectorite/iron oxide nanocomposites	0.36	[56]
	γ-Fe_2_O_3_/SiO_2_/chitosan composite	34.29	[57]
	CoFe_2_O_4_	94.46	This study
	CoFe_2_O_4_–Chit	66.18	This study
	CoFe_2_O_4_ from binary solutions with CR	117.50	This study
	CoFe_2_O_4_–Chit from binary solutions with CR	81.30	This study

**Table 4 nanomaterials-11-00711-t004:** The kinetic parameters for the dyes’ adsorption onto CoFe_2_O_4_ and CoFe_2_O_4_–Chit obtained by nonlinear regression of the experimental data.

Sample	CR-CoFe_2_O_4_	CR-CoFe_2_O_4_-Chit	MO-CoFe_2_O_4_	MO-CoFe_2_O_4_-Chit
Single solutions				
Q_e_ exp (mg/g)	136.13	15.04	92.20	64.50
Pseudo-first-order model				
k_1_ (min^−1^)	0.0497 ± 0.0043	0.0396 ± 0.0049	0.0247 ± 0.0022	0.0301 ± 0.0024
Q_e_ cal (mg/g)	135.69 ± 2.77	14.75 ± 0.44	92.86 ± 2.14	64.59 ± 1.27
R^2^	0.9843	0.9693	0.9868	0.9892
AIC	48.38	7.184	39.82	29.48
Pseudo-second-order model				
k_2_ (10^−3^ g/mg∙min)	0.4731 ± 0.1000	3.2750 ± 0.9000	0.2775 ± 0.1000	0.5131 ± 0.2000
Q_e_ cal (mg/g)	146.55 ± 8.55	16.18 ± 0.94	106.05 ± 8.98	72.47 ± 5.34
R^2^	0.9832	0.9862	0.9782	0.9806
AIC	49.12	−1.60	45.34	35.93
Intraparticle diffusion model				
k_i_	8.89	1.01	-	-
R^2^	0.8296	0.9278	-	-
Binary solutions				
Q_e_ exp (mg/g)	100.50	64.50	23.75	21.25
Pseudo-first-order model				
k_1_ (min^−1^)	0.0899 ± 0.0143	0.0301 ± 0.0024	0.0439 ± 0.0072	0.0471 ± 0.0062
Q_e_ cal (mg/g)	95.39 ± 3.29	64.59 ± 1.27	22.34 ± 0.88	20.27 ± 0.63
R^2^	0.8267	0.9843	0.8994	0.9325
AIC	51.18	29.18	23.10	16.78
Pseudo-second-order model				
k_2_ (10^−3^ g/mg∙min)	1.3470 ± 0.3698	0.5131 ± 0.1754	2.3590 ±1.0000	2.8520 ± 1.0000
Q_e_ cal (mg/g)	101.22 ± 4.90	72.47 ± 5.75	24.51 ± 1.60	22.11 ± 0.92
R^2^	0.9569	0.9682	0.9651	0.9847
AIC	37.28	35.05	11.35	0.73

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
