# Peer review of "Comparative Study of CoFe2O4 Nanoparticles and CoFe2O4-Chitosan Composite for Congo Red and Methyl Orange Removal by Adsorption"

_nanomaterials, 2021, doi:10.3390/nano11030711_

Round 1

Reviewer 1 Report

In this manuscript adsorption of congo red and methyl orange onto cobalt ferrite and cobalt ferrite-chitosan composite was described. Manuscript is really well prepared and brings interesting results and I suggest it for publication after minor revisions.

Comments

  1. The sentence regarding the development of bacteria and viruses (lines 67-69) should be omitted or rewritten. There is no evidence between dye contaminated waters and virus development. Similarly, the sentence in line 70 is not well presented “DNA of cancer-causing cells”. Lines 52 – 78 are based from 3 reviews articles…maybe you should use original sources to properly describe this part of introduction section.
  2. Line 119 – bracket should be added after “Sigma Aldrich”
  3. Adsorbents were synthetized using relatively complex process (cost and energy ineffective). Could such material compete with cheap adsorbents e.g. based on waste biomass?
  4. Why the NH4OH was used to ensure higher pH values?
  5. Standard deviations or errors should be added to all calculated parameters (isotherm and kinetics).
  6. In binary adsorption experiments the simple Freundlich and Langmuir models were applied. However, competitive interactions between dyes and adsorbents are expected during adsorption in binary solutions (sum of Qmax from single systems is higher than sum of Qmax from binary system). Therefore, the use of competitive Langmuir and Freundlich equations would be more appropriate. Please check relevant literature.  
  7. To analyse the kinetic od dye adsorption linear equations of pseudo-first and pseudo-second order models were applied. Considering the fact that isotherm parameters calculations were realized using non-linear regression analysis I recommend authors to use this procedure also for kinetics parameters calculations.
  8. What was the desorption efficiency (%)? This should be at least mentioned in the text.

Reviewer 2 Report

As we also publish in that field, i.e. Water purification, I found this paper excellent. Only minor changes are required before final acceptance.

  • line 76: induce
  • line 136: 15s/step
  • line 137: 30s/step
  • line 158: I could not find in my dictionaries either "thermostatted"or "thermostated". Please, do correct!
  • lines 282-288: FTIR spectrum of chitosan Figure 2: I could not find easily on Fig. 2 the frequencies given in the text.
  • line 333: distinguished.
  • lines 362-363: analysed
  • line 428: analysing
  • line 490: I could not find 3 hours for CR and 4 hours for MO. The figures show shorter times.
  • lines 540-541: ...that the respective model is more probable to characterize the sorption process than the alternative model [43].
  • line 567: Analysing
  • line 590: their choice

Reviewer 3 Report

This study focused on preparation of CoFe2O4 and adsorbent and evaluation of their adsorption capacity toward MO and CR. The prepared absorbents were characterized by FT-IR, XRD, SEM and TEM. Then the factors affecting adsorption behavior including pH, contact time, desorption agents were studied. At optimum condition, the adsorption behaviors were used to fit to thermodynamic analysis (Langmuir/Freundlich isotherm model) and kinetic analysis (pseudo first order and second order model). Finally, the recycling used tests were also discussed. Overall, this study provided important information about adsorption behaviors for both adsorbents. However, some unclear parts were also found. Thus this manuscript was not suggested to accept in current form.

  1. According to manuscript, the interactions between both adsorbents and dyes come from electrostatic interaction and hydrophobic force at low and neutral pH range. What is the major reason making different adsorption ability of dyes for both adsorbents? If possible, this results should be combined with thermodynamic and kinetic analysis.
  2. In this manuscript, how to separate adsorbents and testing solution? Magnetic separation or centrifugation? This information should be provided.
  3. Why select MO and CR as model? How about other dyestuff?
  4. For practicality, environmental water samples such as industrial waste water, tap water, and lake water should be evaluated because matrix effect maybe influence the adsorption behaviors.
  5. According to manuscript, the advantage of CoFe2O4-Chitosan comparing to CoFe2O4 is in column system. Please prove it.

Round 2

Reviewer 3 Report

They provided reasonable responses for the questions in the article. Thus, this article can be accepted in current form.